Diversity and conservation of mammals in indigenous territories of southern Mexico: proposal for an “Archipelago Reserve”

Briones-Salas Miguel 1
Galindo-Aguilar Rosa E. xanatsa@gmail.com 1
González Graciela E. 1
Luna-Krauletz María Delfina 2
1 Centro Interdisciplinario de Investigación para el Desarrollo Integral Regional, Unidad Oaxaca, Instituto Politécnico Nacional , Santa Cruz Xoxocotlan , Oaxaca , México
2 Instituto de Estudios Ambientales, Universidad de la Sierra Juárez , Ixtlán de Juárez , Oaxaca , México
Pavoine Sandrine
Electronic publication date: 2023 Nov 7
Publication date: 2023
Volume: 11
Electronic Location ID: e16345
Received 2023 Mar 23; Accepted 2023 Oct 3
Copyright: ©2023 Briones-Salas et al.
Copyright year: 2023
Copyright holder: Briones-Salas et al.
License: This is an open access article distributed under the terms of the Creative Commons Attribution License, which permits unrestricted use, distribution, reproduction and adaptation in any medium and for any purpose provided that it is properly attributed. For attribution, the original author(s), title, publication source (PeerJ) and either DOI or URL of the article must be cited.
License URL: https://creativecommons.org/licenses/by/4.0/

Keywords: Áreas destinadas voluntariamente a la conservación, Tropical rainforest, Cloud forest, Chinantla, Bats, Sierra Madre de Oaxaca

Funding: Secretary of Research and Graduate Studies of the National Polytechnic Institute SIP: 20220797 Mexican National Commission of Science and Technology (CONACyT) #622396 Commission for the Operation and Promotion of Academic Activities (COFAA) and the Research Incentive Program (EDI) at Instituto Politécnico Nacional The project was supported by the Secretary of Research and Graduate Studies (SIP: 20220797) of the National Polytechnic Institute. CONANP provided information on community monitoring in the region. REGA thanks the scholarship (#622396) granted by the Mexican National Commission of Science and Technology (CONACyT) to pursue her Ph.D. in Sciences in Management and Conservation of Natural Resources at the CIIDIR, Oaxaca, Instituto Politécnico Nacional. MB-S thanks the Commission for the Operation and Promotion of Academic Activities (COFAA) and the Research Incentive Program (EDI) at Instituto Politécnico Nacional for the fellowship, as well as the National System of Researchers (SNI) for its recognition. The funders had no role in study design, data collection and analysis, decision to publish, or preparation of the manuscript.

==============================
Southern Mexico’s tropical forests are home to the country’s highest richness of mammal species; La Chinantla region is situated within this area, its name from the indigenous group residing in the area and holding territorial ownership, namely the Chinantecos. In La Chinantla, there are no Protected Areas; instead, there are Areas Destined Voluntarily for Conservation (ADVC) and “Voluntary Conservation Areas” (VCA), that are managed by local inhabitants through social consensus. These ADVC may function as an archipelago reserve, which represents regional diversity, including the social context, through complementarity. To verify its biodiversity, we analyzed the richness, composition, distribution, and conservation of wild mammals in the region. Records were obtained from four sources—primary data collection, databases, scientific literature, and community monitoring—and were organized into four zones based on altitudinal and vegetation gradients. We compared the diversity between zones for three categories of mammals: small (<100 gr.), bats, and medium and large (>100 gr.). 134 species were identified comprising 11 orders, 26 families and 86 genera. The zone with highest elevation presented the greatest species richness for the assemblage of mammals and terrestrial mammals, while the zone with the lowest elevation had the highest richness of bats. For each mammal category, the zone with the most species also registered the highest number of exclusive species. For the assemblage of mammals and for medium and large mammals, the similarity index was highest between the two intermediate zones, while for small mammals and bats, the greatest similarity occurred between the areas of higher altitude. The study region was found to have the second highest richness of mammals in Mexico. Finally, we suggest that the conservation proposals by indigenous people could function as a set of “islands” that promote the conservation of biodiversity, possibly as an Archipelago Reserve.

Introduction

A recurring pattern in biodiversity is that species richness increases as latitude decreases (Rodhe, 1999), which is one of the reasons why the most biodiverse ecosystems in the world are in equatorial regions. In addition, the mountains in the tropics have high biological diversity, since they contain species with different origins and evolutionary histories along their altitudinal gradient (Graham et al., 2014). Of this biological diversity, beta diversity is most notable, mainly due to geographic isolation and speciation processes (Mastretta-Yanes et al., 2015).

The Sierra Madre del Sur is in southern Mexico, in addition to being composed of temperate vegetation, is home to tropical forests. Species richness is high in these forests, not only in terms of taxonomic diversity but also phylogenetic and functional diversity (Espinosa, Ocegueda-Cruz & Luna-Vega, 2016; Aguilar-Tomasini, Martin & Speed, 2021).

One of the largest humid tropical forests in Mexico is located in a region in the north of the state of Oaxaca called La Chinantla; the region has been inhabited since the pre-Columbian era by the Chinantecs, an ethnic group that maintains the milpa system, which relies on subsistence livestock and hunting in some areas (Legarreta, 2010) and it is considered a priority terrestrial region for conservation in Mexico (Arriaga et al., 2000) and a priority conservation area for Mesoamerica because of its biodiversity and endemic species (De Albuquerque et al., 2015). It is a mountainous area whose lowlands has large areas of tropical rainforest (TRF) and montane cloud forest (MCF), both of which house extremely biodiverse plant communities (Rzedowski, 1996), while its highlands contain pine-oak-forests (P-OF). Being composed mainly of TRF and MCF, this region is an important source of supply of water resources, carbon sequestration, and other environmental services (Galicia & Zarco-Arista, 2014).

Several studies have revealed a great richness of vertebrate species in La Chinantla (Noria-Sánchez, Prisciliano-Vázquez & Patiño Islas, 2015; Briones-Salas, Cortés-Marcial & Lavariega, 2015; Simón-Salvador et al., 2021), particularly of mammals (Pérez-Lustre, Contreras-Diaz & Santos-Moreno, 2006; Pérez-Irineo & Santos-Moreno, 2012; Del Rio-García et al., 2014). Moreover, recent records have highlighted species of paramount importance for conservation efforts, as they are classified as threatened at both national and international levels (except for Caluromys. derbianus): Tapirus Bairdii, Panthera onca, Ateles geoffroyi, and C. derbianus (Lira-Torres et al., 2006; Figel et al., 2009; Ortiz-Martínez et al., 2012; Galindo-Aguilar et al., 2019). The region has also been recognized as a biological corridor for jaguar (P. onca) conservation in southern and northern Mexico (Rodríguez-Soto et al., 2011), and researchers have recently begun suggesting that it could be a viable region for jaguar conservation (Jȩdrzejewski et al., 2018; Lavariega et al., 2020).

Despite the high diversity recorded in the region, La Chinantla does not contain any Protected Areas (PAs)—where the federal or state government exercises jurisdiction. However, Chinantecs have established almost 31 communitarian conservation areas which are volunteered elected for biodiversity conservation, namely “Areas Destined Voluntarily for Conservation” (ADVC, 27) and “Voluntary Conservation Areas” (VCA, 4). The former is federally certified by the National Commission of Protected Natural Areas (CONANP, by its Spanish acronym) while the latter is not. In both cases, these spaces are managed by the inhabitants themselves through social consensus, which establishes rules of use, including restrictions on hunting and looting plant species and removing plant cover for agricultural and livestock activities (Anta-Fonseca & Mondragón-Galicia, 2006; Lele et al., 2010). ADVCs and VCAs cover variable extensions of forest and aim to protect the most fragile natural environments. They are in areas with high biological and cultural diversity and are commonly inhabited by species that fall into some national and international threat category.

The ADVC in La Chinantla are distributed at an altitudinal gradient of 50–2,500 masl and comprise a little more than 58,765.78 ha (CONANP-Chinantla Office), almost half of which is certified by the state of Oaxaca (125,923 ha) (CONANP, 2020).

While that there are no PAs, such as national parks and biosphere reserves, in the region, which would represent a conservation strategy due to their unique and variable-sized areas (Halffter, 2007; Moctezuma, Halffter & Arriaga-Jiménez, 2018), alternative strategies could be considered. One proposed approach is the establishment of small, protected areas to safeguard the entire regional diversity and enhancing beta diversity. This model has been referred to in other studies as the “Archipelago Reserve” (AR) (Halffter, 2007; Moctezuma, Halffter & Arriaga-Jiménez, 2018). The objective of this model is to represent regional diversity, including the social context, through complementarity. The use of multiple areas aims to increase species exchange among remnants in fragmented landscapes or between traditional and rustic agroecosystems with high biological diversity, valuing productive practices and sustainable development (Halffter, 2007; Moctezuma, Halffter & Arriaga-Jiménez, 2018). Given the biological and social significance of La Chinantla and considering the presence of small communitarian conservation areas (ADVC y VCA) geographically distributed within a priority site and encompassing a significant altitudinal gradient in a mountainous zone, we propose that the region may function as an archipelago reserve with substantial potential for biodiversity conservation. Therefore, we establish the following objectives: (1) analyze the distribution of mammals as a focus group along an altitudinal gradient; (2) to determine whether species richness distribution is homogeneous along this gradient or if there is species turnover among different altitudinal steps.

Materials & Methods

Study area

La Chinantla is in northeast Oaxaca, in the foothills of the Sierra Madre de Oaxaca and in the Papaloapan River basin. It comprises 14 municipalities that are in the subregions of Sierra Madre de Oaxaca and Planicie Costera del Golfo (Ortiz-Pérez, Hernandez-Santana & Figueroa-Mah-Eng, 2004) (Fig. 1).

Figure 1 Geographical location and sources of data on wild mammals in La Chinantla region, northeast of Oaxaca, Mexico.

Zone 1 = 4–400, Zone 2 = 400–1,000, Zone 3 = 1,000–1,500, Zone 4 = 1,500–3,000. Altitude is in masl.

The climate in La Chinantla is hot and humid in the lowlands, and cool and humid in the highlands with an average annual temperature of 16 °C and 25 °C, respectively (Meave, Rincón & Romero-Romero, 2006). The region has high humidity and annual rainfall between 3,600 and 5,800 mm. Its altitude ranges from 0 to 3,000 masl, with slopes between 6° and 45° in 80% of the territory (Meave, Rincón & Romero-Romero, 2006). Tropical rainforests (TRF) and montane cloud forests (MCF) predominate (INEGI, 2016).

Dividing the study area into zones

Based on the altitudinal gradient and the types of vegetation, we divided the study region in four zones. Zone 1: 0 to 400 masl and warm climates, lowland TRF dominates, combined with secondary vegetation (SV), agriculture (A), and livestock areas; contains 10 ADVCs. Zone 2: 401 to 1000 masl, with warm and semi-warm climates, TRF in the midlands, sub-deciduous tropical forest (SDTF) combined with SV, A, and livestock areas; contains nine ADVCs. Zone 3: 1,001 to 1,500 masl, with humid subtropical climates, MCF vegetation, mixed with small fragments of highland TRF, SV, and to a lesser extent A; contains four ADVCs. Zone 4: >1,501 masl, with temperate and humid climates and temperate pine-oak forests (P-OF) mixed with small fragments of MCF, SV, and A; contains three VCAs and one ADVC (Fig. 1).

Obtaining records

The records used for this study came from four sources:

1. Primary data collection. The collection of specimens was carried out by four research teams, led by the authors. Approximately 40 visits were made to the study area, around four per year, with an average of three consecutive days each, covering both the rainy and dry seasons, from the year 2010 to 2020. During each visit, efforts were made to cover the widest area both inside and outside the ADVCs, considering all altitudinal gradients present, and changing sampling sites daily.

Conventional sampling techniques were used for different mammal groups: small mammals (<100 g), such as shrews and rodents; bats; and medium to large mammals (>101 g) (Medellín, 1994). During each visit, small mammals were captured using an average of 100 Sherman traps (7.6 × 8.9 × 22.5 cm) baited with a mixture of peanut butter, vanilla extract, and oats. The traps were set daily along two 500 m linear transects, either within the vegetation or near water bodies, with a 10 m spacing between each trap. The sampling effort was 2,171 trap-nights per year of work. Additionally, 100 pitfall traps were placed during each sampling period at 2 m intervals in areas with leaf litter and near fallen logs, with a depth of 30 cm. For both cases, the traps were set in the afternoon and checked the following morning.

Bats were captured using mist nets (12 × 2.6 m). During each visit, an average of three nets were set up in locations near water bodies or within the vegetation. The nets remained open for eight hours each night, over three consecutive days. The sampling effort per site was calculated by multiplying the length and width of the mist nets by the number of hours they were open, in addition to the number of nights and the number of nets used. The result was expressed as m2 net/hour based on the method proposed by Medellín (1994). The total sampling effort per year was 1,123 m of net in 96 h, divided into 12 nights sampling. Thus, the sampling effort was 26,880 m of net per night.

For small mammals and bats, once captured, somatic measurements, reproductive condition, weight, sex, and age data were obtained. Most of the specimens were released at the same capture site, except for a minimum number of individuals that were collected and prepared as museum specimens using taxidermy techniques following the recommendations of Hall (1981). Subsequently, they were deposited in the Mastozoological Collection of the Centro Interdisciplinario de Investigación para el Desarrollo Integral Regional, Unidad Oaxaca (OAX.MA.026.0497). Specimens were collected under a scientific collection permit issued by the Mexican Ministry of Environment and Natural Resources (FAUT-0037; SEMARNAT).

For medium to large mammals, two randomly distributed linear transects of approximately 2.5 km in length were traversed in each locality to search for signs (tracks and feces) during each visit. The transects were walked daily by two observers. Once signs were located, data on tracks was collected (geographic location, length and width measurements), photographs were taken, and plaster casts were made. To complete the inventory, four Tomahawk-type traps with double folding doors (24 × 6 × 6 inches) were used, baited with ripe fruits and sardines. The traps were placed within the vegetation and near water bodies on the first day of work and checked daily in the morning. In case of any capture, photographs of the individual were taken, and data on species, sex, age, and if possible, somatic measurements was recorded. All specimens were released at the same capture site.

For all collection sites, geographic coordinates and elevation were obtained using a Global Positioning System (GPS) with the WGS84 datum.

2. Databases. We obtained records of wild mammals in museums and scientific collections; five domestic: la Colección Nacional de Mamíferos (CNMA), la Colección de la Escuela Nacional de Ciencias Biológicas del IPN (ENCB), la Colección del Museo de Zoología de la Facultad de Ciencias de la UNAM (MZFC), la Colección de la Universidad Autónoma Metropolitana-Iztapalapa(UAMI) y la Colección del Centro Interdisciplinario de Investigación para el Desarrollo Integral Regional, Unidad Oaxaca (CIIDIR-OAX.), IPN (OAXMA). The foreign collections were ten: Field Museum Natural History (FMNH), American Museum of Natural History(AMNH), Texas A&M University (TWWC), Angeles County Museum (LACM), Museum of Zoology, University of Michigan (UMMZ), Texas Tech University(TTU), University of New Mexico, Museum of Southwestern Biology (MSB), Carnie Museum of Natural History (CM), University of Florida, Florida Museum of Natural History (UF) y Natural History Museum, Kansas University (KU). Data were also obtained from the Global Biodiversity Information Facility (GBIF) portal.

3. Literature review. A review of the scientific literature was carried out in databases including Scopus, Scielo, Redalyc, Google Scholar, and Elsevier for the years 1969–2022. We use the keywords “mammals”, “mamíferos” or “Chinantla”. Topics such as distribution, new records of species, and expansion of taxa distribution areas were reviewed. Each scientific article discovered underwent a meticulous review to ascertain the presence of the required information. After this evaluation, the ensuing data was extracted: municipality, type of vegetation, and species.

4. Community monitoring. The community monitoring was done by the local inhabitants. The authorities, as well as the residents of each community where the camera traps were deployed, provided their endorsement to the National Commission of Protected Natural Areas (CONANP). We formally request the information that has been generated through community monitoring within the Chinantla ADVCs from CONANP (DRFSIPS-0095-2019). We have carefully reviewed all the photos and videos captured during the monitoring conducted between 2011 and 2014, with the aim of creating a database containing independent events (see details below). Community monitors placed 5 camera trap models (Bushnell. n = 97; Moultrie; n = 26; Wildview, n = 3; Ltl Acorn, n = 2; and Stealth Cam, n = 1). The cameras were programmed to operate 24 h a day and to capture photos (1–5 photos) and/or videos (10–30 s long). The traps were placed on trees or stakes 10–40 cm above the ground, generally one meter from the roads where the monitors had observed fauna or tracks. We use for the analysis only the independent events: (1) consecutive photographs of different individuals of the same or different species, (2) consecutive photographs of individuals of the same species taken more than 24 h apart, (3) nonconsecutive photos of individuals of the same species (O’Brien, Kinnaird & Wibisono, 2003). Our analysis used data from 129 camera trapping stations over a four-year period (2011–2014) in 18 indigenous communities in the region. The total sampling effort was of 4,373 trap-nights; 2,257 in Zone 1 (61 camera traps), 1,354 in Zone 2 (45 camera traps), 540 in Zone 3 (19 camera traps) and 222 in Zone 4 (four camera traps). The details of dates and time in which the camera traps in the field can be observed in Fig. S1.

Data analysis

The collected specimens were identified using specialized guides (Ceballos & Oliva, 2005; Aranda-Sánchez, 2012; Álvarez Castañeda, Álvarez & Gonzáles-Ruiz, 2017). The nomenclature was updated following Ramírez-Pulido et al. (2014), with some recent modifications. Specialized literature (Hall, 1981; Ceballos & Oliva, 2005; Carraway, 2007; Briones-Salas, Cortés-Marcial & Lavariega, 2015) was consulted to identify taxa endemic to Mexico and Oaxaca.

Species richness was counted as the total number of species recorded in the entire region and the four zones. We created species accumulation curves for species richness based on rarefaction and extrapolation (Chao & Jost, 2012) using the program iNEXT (Chao, Ma & Hsieh, 2022).

Beta diversity was obtained using the Jaccard qualitative similarity index; this index compares species communities between two sites to determine which species are shared and which are distinct. Using this measure, we assess the disssimilarity between pairs of communities for all mammals, small mammals, bats, and medium to large mammals from the four sites and ranged from 0 (shared species between two sites) and 1(sites do not have the same composition). We measured total β diversity and the turnover and nestedness components between zones based on the Jaccard index (Baselga et al., 2018). We calculated: (1) total β diversity (βju), (2) turnover β diversity (βtu), and (3) nestedness β diversity (βne). These parameters were computed using the betapart package (Baselga et al., 2018) in R (version 3.3.3; R Core Team, 2017), utilizing the Incidence-based pair-wise dissimilarities function, beta.pair. Additionally, we employed the unweighted pair group method for arithmetic averages (UPGMA). The analyzes were performed in the program PAST.

Conservation and protection statuses for the species were recorded based on the IUCN Red List (IUCN, 2022), the appendices of the CITES (CITES, 2013), and NOM-059-SEMARNAT-2010 (SEMARNAT, 2010).

Table 1 Terrestrial mammals recorded in La Chinantla Oaxaca, Mexico.

Taxonomic category	Size	Record type	Zone	Vegetation type	Year	NOM	IUCN	CITES	
Orden Didelphimorphia									
Familia Didelphidae									
Caluromys derbianus (Waterhouse, 1841)	M-L	CM	1	TRF	2012	A	LC		
Didelphis marsupialis Linnaeus, 1758	M-L	C, D, L	1, 2, 3	MCF, SDTF, TRF	1964–2015		LC		
Didelphis virginiana Kerr, 1792	M-L	C, D, L	1, 2, 3	MCF, P-OF, SDTF, TRF	1901–2016		LC		
Marmosa mexicana Merriam, 1897	S	C, D	2, 3, 4	MCF, P-OF, TRF	1962–2005		LC		
Philander opossum (Linnaeus, 1758)	M-L	CM, D	1, 2, 3	MCF, SDTF, TRF	1962–2014		LC		
									
Orden Cingulata									
Familia Dasypodidae									
Dasypus novemcinctus Linnaeus, 1758	M-L	C, CM, L	1, 2, 3	MCF, SDTF, TRF	2005–2015		LC		
									
Orden Pilosa									
Familia Myrmecophagidae									
Tamandua mexicana (de Saussure, 1860)	M-L	CM, D, L	1, 2, 3	MCF, SDTF, TRF	1990–2014	P	LC		
									
Orden Eulipotyphla									
Familia Soricidae									
Cryptotis berlandieri (Baird, 1858)	S	C	4	P-OF	2009	Pr	LC		
Cryptotis goldmani (Merriam, 1895) MX	S	D	4	TRF	1972	Pr	LC		
Cryptotis magnus (Merriam, 1895) OAX	S	D	1, 2, 3, 4	MCF, P-OF, TRF	1959–1991	Pr	VU		
Cryptotis mexicanus Coues, 1877 MX	S	D, C	3, 4	MCF, P-OF	1964–2009		LC		
Sorex macrodon Merriam, 1895	S	D	3	MCF	1969–1975	A	VU		
Sorex saussurei Merriam, 1892	S	D	3, 4	MCF	1964–1986		LC		
Sorex ventralis Merriam, 1895	S	D	3	MCF	1995		LC		
Sorex veraecrucis Jackson, 1925	S	D, C	3	MCF, P-OF, TRF	1964–1993	A	LC		
Sorex veraepacis Alston, 1877	S	D, C	3, 4	MCF, P-OF, TRF	1964–2005	A	LC		
Orden Chiroptera									
Familia Emballonuridae									
Balantiopteryx io Thomas, 1904	B	D, L	1	SDTF. TRF	1962–2006		VU		
Balantiopteryx plicata Peters, 1867	B	C, D, L	1, 4	P-OF, SDTF	1962–2009		LC		
Diclidurus albus Wied-Neuwied, 1820	B	C	1	SDTF	2014		LC		
Peropteryx macrotis (J.A. Wagner, 1843)	B	D	1	SDTF, TRF	1962–1988		LC		
Rhynchonycteris naso (Wied-Neuwied, 1820)	B	D	1	SDTR	1990	Pr	LC		
Saccopteryx bilineata (Temminck, 1838)	B	D, C	1,4	P-OF, SDTR, TRF	1962–2014		LC		
Familia Molossidae									
Molossus aztecus de Saussure, 1860	B	D	2	SDTR, TRF	1962		LC		
Molossus rufus E.Geoffroy Saint-Hilaire, 1805	B	D	1, 2	SDTR, TRF	1960–1969		LC		
Tadarida brasiliensis (I. Geoffroy Saint- Hilaire, 1824)	B	D	4	P-OF	1988		LC		
Familia Mormoopidae									
Mormoops megalophylla (Peters, 1864)	B	D	1, 3	P-OF, SDTF. TRF	1962–1988		LC		
Pteronotus fulvus Thomas, 1892	B	D	1	TRF	1988		LC		
Pteronotus mesoamericanus Smith, 1972	B	C, D	1, 4	P-OF, SDTF, TRF	1960–2009		LC		
Pteronotus psilotis (Dobson, 1878)	B	D	1	SDTF	1969		LC		
Familia Phyllostomidae									
Anoura geoffroyi Gray, 1838	B	D, C, L	2, 3, 4	MCF. P-OF, TRF	1964–2009		LC		
Artibeus jamaicensis Leach, 1821	B	D, C, L	1, 2, 3, 4	MCF, P-OF, SDTF, TRF	1960–2014		LC		
Artibeus lituratus (Olfers, 1818)	B	D, C	1, 3, 4	MCF, P-OF, SDTF, TRF	1962–2009		LC		
Carollia perspicillata (Linnaeus, 1758)	B	D, C	1, 2, 3	MCF, SDTF, TRF,	1960–2014		LC		
Carollia sowelli R.J. Baker, Solary y Hoffmann, 2002	B	D, L	1, 2, 3, 4	MCF, P-OF, SDTF, TRF	1962–2006		LC		
Carollia subrufa (Hahn, 1905)	B	D, C,	1, 4	P-OF, SDTF, TRF	1969–2014		LC		
Centurio senex Gray, 1842	B	D, C, L	1, 3, 4	P-OF, SDTF, TRF	1960–2014		LC		
Choeroniscus godmani (Thomas, 1903)	B	C	4	P-OF	2009		LC		
Chiroderma villosum Peters, 1860	B	D	1	TRF	2001		LC		
Chrotopterus auritus (Peters, 1856)	B	D	2	TRF	1966		LC		
Dermanura phaeotis Miller, 1902	B	D, C	1, 3	MCF, TRF, SDTF	1962–2014		LC		
Dermanura tolteca (Saussure, 1860)	B	D, L	1, 2, 3, 4	MCF, P-OF, SDTF, TRF	1964–2006		LC		
Desmodus rotundus (E. Geoffroy Saint-Hilaire, 1810)	B	D, C, L	1, 3, 4	P-OF, SDTF, TRF	1960–2014		LC		
Enchisthenes hartii (Thomas, 1892)	B	L	1, 2	MCF, SDTF	2005–2006	Pr	LC		
Glossophaga commissarisi Gardner, 1962	B	D, C	1, 2, 4	P-OF, SDTF, TRF	1962–2009		LC		
Glossophaga leachii (Gray, 1844)	B	D, C, L	1, 4	P-OF, SDTF, TRF	1969–2009		LC		
Glossophaga morenoi Martínez y Villa, 1938 MX	B	C	4	P-OF	2009		LC		
Glossophaga soricina (Pallas, 1766)	B	D, C, L	1, 2, 3, 4	MCF, P-OF, SDTF, TRF	1962–2014		LC		
Hylonycteris underwoodi Thomas, 1903	B	D	1,3	MCF, P-OF, SDTF, TRF	1962–1981		LC		
Leptonycteris yerbabuenae* Martínez y Villa, 1940	B	D, C	1, 4	P-OF, SDTF	1962–2009		NT		
Micronycteris microtis Miller, 1898	B	D	1	SDTF	1962–1969		LC		
Mimon cozumelae Goldman, 1914	B	L	1	SDTF	2006	A	LC		
Platyrrhinus helleri (Peters, 1866)	B	D	1, 3	SDTF, TRF	1962–1988		LC		
Phyllostomus discolor (J.A. Wagner, 1843)	B	D, L	1, 2	MCF, SDTF	1974–2006		LC		
Sturnira hondurensis Goodwin, 1940	B	D, C, L	2, 3, 4	MCF, P-OF, TRF	1960–2009		LC		
Sturnira parvidens Goldman, 1917	B	D, C, L	1, 2, 3, 4	MCF, P-OF, SDTF, TRF	1960–2014		LC		
Trachops cirrhosus (Spix, 1823)	B	D	2	TRF	1989		LC		
Uroderma bilobatum Peters, 1866	B	D, C	1	SDTF	1988–2014		LC		
Vampyressa thyone Thomas, 1909	B	L	1	SDTF	2006		LC		
Vampyrodes major G.M.Allen, 1908 (Thomas, 1889)	B	D, L	1, 3	SDTF, TRF	1961–2006		LC		
Vampyrum spectrum (Linnaeus, 1758)	B	D, L	1	SDTF, TRF	2005–2006	P	NT		
Familia Vespertilionidae									
Eptesicus furinalis (D’Orbigny y Gervais, 1847)	B	D, C	1	SDTF, TRF	1962–2014		LC		
Aeroestes cinereus (Palisot de Beauvois, 1796)	B	C	3	P-OF	1999		LC		
Dasypterus intermedius H. Allen, 1862	B	D	1	TRF	1962–1969		LC		
Dasypterus xanthinus (Thomas, 1897)	B	D	1	TRF	1962		LC		
Myotis keaysi J. A. Allen, 1914	B	D	1, 3	MCF, P-OF, SDTF, TRF	1962–1981		LC		
Myotis nigricans (Schinz, 1821)	B	D, C	1, 3, 4	MCF, P-OF, SDTF	1969–2014		LC		
Myotis volans (H.Allen, 1866)	B	D	1	TRF	1960		LC		
Perimyotis subflavus (F. Cuvier, 1832)	B	D	1	TRF	1983		VU		
Orden Primates									
Familia Atelidae									
Ateles geoffroyi* Kuhl, 1820	M-L	D, L	1, 3	MCF, SDTF, TRF	2006–2012	P	EN		
Orden Lagomorpha									
Familia Leporidae									
Sylvilagus gabbi (Linnaeus, 1758)	M-L	D	1	TRF	1964		EN		
Sylvilagus floridanus (J. A. Allen, 1890)	M-L	C, L	1, 4	MCF, P-OF, SDTF	2010–2015		LC		
Orden Rodentia									
Familia Sciuridae									
Sciurus aureogaster F. Cuvier, 1829	M-L	C, CM, D, L	1, 2, 3, 4	MCF, P-OF, SDTF, TRF,	1962–2015		LC		
Sciurus deppei Peters, 1863	M-L	C, CM, D	1, 2, 3, 4	MCF, P-OF, TRF	1964–2015		LC		
Familia Geomyidae									
Heterogeomys hispidus (Le Conte, 1852)	M-L	L	2	SDTF	2011		LC		
Orthogeomys grandis (Thomas, 1893)	M-L	D	4	P-OF	1969–1981		LC		
Familia Heteromyidae									
Heteromys desmarestianus Gray, 1868	S	D, C, L	1, 2, 3, 4	MCF, P-OF, SDTF, TRF	1962–2006		LC		
Heteromys irroratus (Gray, 1868)	S	C, L	1, 3, 4	P-OF, SDTF, A	2006–2018		LC		
Familia Erethizontidae									
Coendou mexicanus Kerr, 1792	M-L	C, D, L	1, 2	SDTF, TRF	1992–2015	A	LC		
Familia Dasyproctidae									
Dasyprocta mexicana de Saussure, 1860 MX	M-L	C, CM, D, L	1, 2, 3	A, MCF, SDTF, TRF	1947–2015		CR		
Familia Cuniculidae									
Cuniculus paca (Linnaeus, 1766)	M-L	C, CM, D, L	1, 2, 3	MCF, P-OF, SDTF, TRF	1964–2015		LC		
Familia Cricetidae									
Habromys chinanteco (Robertson y Musser, 1976) OAX	S	C, D	4	MCF, P-OF	1970–2006		CR		
Habromys ixtlani (Goodwin, 1964) OAX	S	C, D	3, 4	MCF, P-OF	1964–2006		CR		
Habromys lepturus (Merriam, 1898) OAX	S	D	3, 4	MCF, P-OF	1964–1989		CR		
Handleyomys alfaroi (J. A. Allen, 1891)	S	D	2, 3, 4	MCF, P-OF, TRF	1962–1988		LC		
Handleyomys chapmani Thomas, 1898 MX	S	C, D, L	1, 2, 3, 4	A, MCF, PAS, P-OF, SDTF, TRF	1901–2018		VU		
Handleyomys melanotis Thomas, 1893	S	D	1, 3	MCF	1962–1972		LC		
Handleyomys rostratus Merriam, 1901	S	D	1, 3	P-OK, TRF	1962		LC		
Megadontomys cryophilus (Musser, 1964) OAX	S	C, D	2, 3, 4	MCF, P-OF	1964–2006		EN		
Microtus mexicanus (Saussure, 1861)	S	C, D	3, 4	P-OF	1962–1999		LC		
Microtus oaxacensis Goodwin, 1966 OAX	S	C, D	3, 4	MCF, P-OF	1964–2006		EN		
Nyctomys sumichrasti (de Saussure, 1860)	S	D	1, 3, 4	MCF, P-OF, SDTF, TRF	1962–1991		LC		
Oligoryzomys fulvescens (de Saussure, 1860)	S	C, D, L	1, 2, 3	MCF, P-OF, TRF	1962–2005		LC		
Oryzomys couesi (Alston, 1877)	S	D	1, 2, 3, 4	MCF, P-OF, SDTF, TRF	1961–1989		LC		
Peromyscus aztecus (de Saussure, 1860)	S	C, D, L	2, 3, 4	A, MCF, P-OF	1959–2018		LC		
Peromyscus beatae Thomas, 1903 MX	S	C, D	3, 4	A, P-OF	2018		LC		
Peromyscus furvus J. A. Allen y Chapman, 1897 MX	S	D	3	MCF, P-OF	1974		DD		
Peromyscus gratus Merriam, 1898	S	C	4	P-OF	1999–2005		LC		
Peromyscus levipes Merriam, 1898 MX	S	D, C	3, 4	MCF, P-OF	1963–2005		LC		
Peromyscus labecula (Wagner, 1845)	S	D	3	P-OF	1964		LC		
Peromyscus melanocarpus Osgood, 1904 OAX	S	C, D	2, 3, 4	MCF, P-OF, TRF	1959–2015		EN		
Peromyscus mexicanus (de Sausure, 1860)	S	C, D, L	1, 2, 3, 4	MCF, P-OF, SDTF, TRF	1962–2015		LC		
Reithrodontomys fulvescens J.A. Allen, 1894	S	D	2, 4	P-OF, TRF	1964–1969		LC		
Reithrodontomys mexicanus (de Saussure, 1860)	S	C, D	2, 3, 4	MCF, P-OF, TRF	1962–2015		LC		
Reithrodontomys microdon Merriam, 1901	S	C, D	3, 4	A, MCF, P-OF	1962–2018		LC		
Reithrodontomys sumichrasti (de Saussure, 1860)	S	C, D	3, 4	MCF, P-OF, TFR	1968–1999		LC		
Sigmodon hispidus Say and Ord, 1825	S	D, L	1, 2, 3	MCF, P-OF, SDTF, TRF	1964–2006		LC		
Sigmodon mascotensis J.A. Allen, 1897	S	D	1	TRF	1987–1988		LC		
Sigmodon toltecus (de Saussure, 1860)	S	D	1, 3	P-OF, TRF			LC		
Tylomys nudicaudus (Peters, 1866)	S	D, L	1, 2, 3	MCF, P-OF, SDTF, TRF	1964–2006		LC		
Orden Carnívora									
Familia Felidae									
Puma yagouaroundi (E. Geoffroy Saint-Hilaire, 1803)	M-L	CM, D, L	1, 2, 3, 4	MCF, P-OF, SDTF, TRF	1962–2014	A	LC	I	
Leopardus pardalis (Linnaeus, 1758)	M-L	C, CM, L	1, 2, 3, 4	MCF, P-OF, SDTF, TRF	2010–2015	P	LC	I	
Leopardus wiedii (Schinz, 1821)	M-L	C, CM, L	1, 2, 3	MCF, P-OF, SDTF, TRF	2005–2016	P	NT	I	
Lynx rufus (Schreber, 1777)	M-L	L	4	P-OF	2010		LC		
Herpailurus concolor (Linnaeus, 1771)	M-L	C, CM, L	1, 2, 3, 4	MCF, P-OF, TRF	2003–2015		LC	II	
Panthera onca* (Linnaeus, 1758)	M-L	C, CM, D, L	1, 2, 3	MCF, P-OF, SDTF, TRF	1998–2014	P	NT	II	
Familia Canidae									
Canis latrans Say,1822	M-L	CM, C, L	1, 4	MCF, P-OF, TRF	2013–2015		LC		
Urocyon cinereoargenteus (Schreber, 1775)	M-L	C, L	2, 3, 4	MCF, P-OF, TRF	2003–2015		LC		
Familia Mephitidae									
Conepatus leuconotus (Lichtenstein, 1832)	M-L	L	2	SDTF	2011		LC		
Conepatus semistriatus (Boddaert, 1785)	M-L	CM, L	1, 2, 3, 4	P-OF, TRF	2010–2014		LC		
Mephitis macroura Lichtenstein, 1832	M-L	C, CM	1, 3, 4	MCF, P-OF, SDTF, TRF	2003–2015		LC		
Spilogale angustifrons A. H. Howell, 1902	M-L	C	4	P-OF	1999		LC		
Familia Mustelidae									
Eira barbara (Linnaeus, 1758)	M-L	CM, C, D, L	1, 2, 3	MCF, P-OF, SDTF, TRF	1965–2016	P	LC		
Galictis vittata (Schreber, 1776)	M-L	CM	1	TRF	2013–2014	A	LC		
Lontra longicaudis* (Olfers, 1818)	M-L	C	1	MCF,	2009	A	NT	I	
Mustela frenata Lichtenstein, 1831	M-L	C, CM, D	1, 2, 4	P-OF, TRF	1961–2014		LC		
Familia Procyonidae									
Bassariscus astutus (Lichtenstein, 1830)	M-L	C, L	4	MCF, P -OF	2010–2015	A	LC		
Bassariscus sumichrasti (de Saussure, 1860)	M-L	CM, D	3	MCF, P -OF	1969–2014	Pr	LC		
Nasua narica (Linnaeus, 1766)	M-L	C, CM, L	1, 2, 3	MCF, P-OF, SDTF, TRF	2005–2016	A	LC		
Potos flavus (Schreber, 1774)	M-L	CM, D, L	1, 2, 3	P-OF, TRF, SDTF	1964–2012	Pr	LC		
Procyon lotor (Linnaeus, 1758)	M-L	CM, C, D, L	1, 2, 3, 4	P-OF, TRF, SDTF	1964–2014		LC		
Orden Artiodactyla									
Familia Tayassuidae									
Pecari tajacu (Linnaeus, 1758)	M-L	C, CM, L	1, 2, 3, 4	MCF, P-OF, SDTF, TRF	2005–2015		LC		
Familia Cervidae									
Mazama temama (Kerr, 1792)	M-L	CM, C, D, L	1, 2, 3, 4	MCF, P-OF, SDTF, TRF	1969–2016		DD		
Odocoileus virginianus* (Zimmermann, 1780)	M-L	C	2, 4	MCF, P-OF, TRF	2015		LC		
Orden Perissodactyla									
Familia Tapiridae									
Tapirus bairdii* (Gill, 1865)	M-L	L	1	MCF	2004	P	EN	I	
Notes.

Keys: Size/class: S, small; B, bats; M-L, medium-large; Record type: D, Database; C, Collection; L, Literature; CM, Community monitoring; Zone 1, 0–400 masl; Zone 2, 401–1,000 masl; Zone 3, 1,001–1,500 masl; Zone 4, >1,501 masl; Vegetation: MCF, Montane cloud forest; TRF, Tropical raiforest; P-OF, Pine-oak forest; SDTF, Sub-deciduous tropical forest; A, Agriculture; PAS, Pastures; Year, Year of first and last documented record; Conservation Status: IUCN: CR, Critically Endangered; EN, Endangered; VU, Vulnerable; NT, Near Threatened; EW, Extinct in the wild. CITES: I, II, III. NOM (Official Mexican Standard NOM-059-SEMARNAT-2010): E, Probably extinct in the wild; P, Endangered; A, Threatened; PR, Subject to special protection; Endemism: MX, Endemic to Mexico; OAX, Endemic to Oaxaca.

* Priority species for conservation present in Oaxaca are marked with an asterisk (*) according to SEMARNAT (2014).

Results

Species richness

A total of 134 species were recorded for the entire region, comprising 11 orders, 26 families, and 86 genera, representing 62%, 100%, 89.6%, and 72.8%, respectively, of all the mammals in the state of Oaxaca (Table 1). The orders with the highest number of species were Chiroptera (n = 52) and Rodentia (n = 38). Only one species was recorded in each of the following orders: Cingulata, Pilosa, Primates, and Perissodactyla (Table 1). By categories, small mammals accounted for 41 species, bats for 52, and medium and large for 41. Seventy-three species were recorded during collection and 26 through community monitoring; 103 and 54 species records were recovered from databases and literature search (Alfaro, García-García & Santos-Moreno, 2006; Ibarra et al., 2011; Lira-Torres et al., 2006; Ortiz-Martínez et al., 2012; Perez-Lustre, Contreras-Diaz & Santos-Moreno, 2006; Del Rio-García et al., 2014), respectively (Table 1; Fig. 2). In terms of the types of vegetation cover, the highest species richness was recorded in the Pine Oakforest (n = 88), followed by the Tropical Rainforest (n = 85; Table 1; Fig. 2). For small mammals, five families were recorded—Cricetidae had the most species (n = 29), while Geomyidae and Heteromyidae had fewer (two species each). Five families of bats were recorded, with Phyllostomidae presenting the highest number of species (n = 31) and Molossidae the lowest(n = 3). The medium and large mammals group contained 17 families—Felidae had most species (n = 6; Table 1).

Figure 2 Frequency of records of mammals (small, medium and large and bats) according to Zone, record type and vegetation in La Chinantla, Oaxaca, Mexico.

Zones: Z1, Zona 1; Z2, Zona 2; Z3, Zona 3; Z4, Zona 4; Record type: D, Database; C, Collection; L, Literature; CM, Community monitoring; Vegetation: MCF, Montane cloud forest; TRF, Tropical raiforest; P-OF, Pine-oak forest; SDTF, Sub-deciduous tropical forest; A, Agriculture; Pas, Pastures.

A higher species richness was recorded in Zone 1 for the assemblage of mammals (n = 89), bats (n = 43), and medium to large mammals (n = 32). However, in the case of small mammals, species richness was Zone 3 (n = 35 and Zone 4; n = 29) (Figs. 3 and 4, Table 2).

Figure 3 Rarefaction/Extrapolation curves of mammal species richness in regions of La Chinantla, Oaxaca.

Z1, Zone 1; Z2, Zone 2; Z3, Zone 3; Z4, Zone 4.

Figure 4 Rarefaction/Extrapolation curves for (A) Small mammals (B) Bats and (C) Medium-Large mammals in La Chinantla, Oaxaca.

Z1, Zone 1; Z2, Zone 2; Z3, Zone 3; Z4, Zone 4.

Table 2 Beta diversity total and turnover and nestedness components of the different groups of wild mammals in the four zones of La Chinantla, Oaxaca.

The values in bold indicate the total number of species in each zone, the number of exclusive species is noted in brackets. The Jaccard dissimilarity index values are italicized, and below it, in normal font, is the number of species shared between each pair of zones.

	Zone 1	Zone 2	Zone 3	Zone 4	
All mammals beta.total				
Zone 1	89 (19)	0.61	0.55	0.70	
Zone 2	41	57 (3)	0.52	0.66	
Zone 3	52	44	79 (7)	0.55	
Zone 4	36	32	46	69 (11)	
All mammals beta.turnover				
Zone 2	0.44				
Zone 3	0.51	0.37			
Zone 4	0.65	0.61	0.50		
All mammals beta.nestedness			
Zone 2	0.17				
Zone 3	0.04	0.15			
Zone 4	0.06	0.05	0.05		
Small mammals beta.total	
Zone 1	14 (1)	0.64	0.64	0.81	
Zone 2	8	15 (0)	0.62	0.65	
Zone 3	13	14	35 (5)	0.41	
Zone 4	7	12	24	29 (4)	
Small mammals beta.turnover			
Zone 2	0.60				
Zone 3	0.13	0.22			
Zone 4	0.67	0.40	0.33		
Small mammals beta.nestedness			
Zone 2	0.04				
Zone 3	0.51	0.40			
Zone 4	0.14	0.25	0.08		
Bats beta.total					
Zone 1	43 (17)	0.79	0.65	0.67	
Zone 2	10	15 (2)	0.69	0.71	
Zone 3	16	8	19 (1)	0.62	
Zone 4	16	8	11	21 (3)	
Bats beta.turnover				
Zone 2	0.50				
Zone 3	0.27	0.64			
Zone 4	0.38	0.64	0.59		
Bats beta.nestedness				
Zone 2	0.29				
Zone 3	0.38	0.06			
Zone 4	0.28	0.08	0.03		
Medium–large mammals beta.total			
Zone 1	32 (5)	0.34	0.32	0.65	
Zone 2	23	27 (2)	0.24	0.63	
Zone 3	23	22	25 (1)	0.66	
Zone 4	13	12	11	19 (4)	
Medium–large mammals beta.turnover			
Zone 2	0.21				
Zone 3	0.15	0.21			
Zone 4	0.43	0.50	0.56		
Medium–large mammals beta.nestedness			
Zone 2	0.14				
Zone 3	0.17	0.03			
Zone 4	0.21	0.13	0.10		

For all mammal groups, species extrapolation curves suggest that Zone 1 and 3 have similar species diversity, higher than Zone 2 and 4, although, between Zone 1 and Zone 4, confidence intervals do not overlap (Fig. 3). For small mammals, the zones with higher altitude (Zone 3 and Zone 4) showed the highest species richness, while the lower altitude zones (Zone 1 and Zone 2) were more similar in their composition (Fig. 4A). For bats, the highest species diversity was recorded in the lower zone (Zone 1, n = 43), while the diversity was lower in higher zones, with overlap was only between Zone 1 and Zone 4, unlike to Zone 2 and Zone 3, that it was statistically different (Fig. 4B). The curves for medium to large mammals suggest that the lower altitude zones (Zone 1 and Zone 2) are more similar and have higher diversity, while the higher altitude zones (Zone 3 and Zone 4) show lower species diversity, although, all confidence intervals overlap (Fig. 4C). It is worth noting that neither the diversity interpolation nor the extrapolation curves reached an asymptote for any of the four groups in any of the zones, indicating the need for additional sampling (Figs. 3 and 4).

Beta diversity

Considering all mammal species, 18 were shared across the four zones (14%). In Zone 1, 19 species were exclusive, while in Zone 2, only three species were exclusive. Zones 1 and 4 had the highest degree of species dissimilarity (DI = 0.70; Table 2). For small mammals, five were shared across the four zones. In Zone 3, five species were exclusive, while Zone 2 had no exclusive species. The highest dissimilarity index for small mammals was observed between Zones 1 and 4 (DI = 0.81). For the bat, five species were shared across the four zones. Zone 1 recorded the highest number of exclusive species (n = 17), while only one exclusive species was recorded in Zone 3. Zones 1 and 2 presented the highest value of dissimilarity (SI = 0.79). Finally for the medium to large mammal, eight species were shared among the four zones. In the lower zone (Zone 1), five exclusive species were recorded, while in Zone 3, only one exclusive species was registered. Zones 3 and 4 exhibited the highest dissimilarity index (DI = 0.66; Table 2). In general terms, it was observed that the species turnover component had a much greater impact on the total beta value for the four groups compared to the nestedness component.

For the assemblage of mammals and small mammals, the similarity dendrograms showed two groups, one for species from the highest altitude zone (Zone 4) and the other containing the remaining zones. For bats, species from Zone 2 are separated from the other zones. For medium to large mammals, species from Zones 2, 3, and 4 form a separate group from those in the lower altitude zone (Zone 1) (Fig. 5).

Figure 5 Jaccard dissimilarity among mammal assemblages considering all species, bats, small mammals, and medium and large-sized mammals in La Chinantla, Oaxaca.

State of conservation

Twenty-four species are listed in NOM-059 (SEMARNAT, 2010), 18 in the list of the International Union for Conservation of Nature (IUCN), and seven in CITES (Table 1). The first two lists shared seven species: Cryptotis magnus (Zone 1–4), Vampyrum spectrum (Zone 1), Ateles geoffroyi (Zone 1 and 3), Leopardus wiedii (Zone 1, 2 and 3), Panthera onca (Zone 1, 2 and 3), Lontra longicaudis (Zone 1), and Tapirus bairdii (Zone 1) (Table 1).

Discussion

Species richness and diversity

Our results highlight the value of indigenous lands for mammal conservation, specifically through community conservation areas. Despite the absence of formal protected natural areas in the region, the mammalian richness in this Mexican region remains remarkably high. La Chinantla ranks as the second location in Mexico with the highest mammalian richness (n = 134), below La Selva Zoque, which is located across three states in southern Mexico: Oaxaca, Chiapas, and Veracruz (n = 149) (Lira-Torres, Galindo-Leal & Briones-Salas, 2012). Please note that La Selva Zoque (∼1,000,000 ha) is considerably larger than La Chinantla (∼58,765.78 ha)(CONANP-Oficina Chinantla). Nevertheless, the recorded species count is nearly identical, underscoring the significance of indigenous territories for mammalian conservation. It is important to point out that mammalian richness in La Chinantla is greater than at other highly biodiverse sites in Mexico, such as La Selva Lacandona, Chiapas (n = 125) (March & Aranda, 1992); La Reserva de la Biósfera el Triunfo, Chiapas (n = 112) (Espinoza-Medinilla, Anzures Dadda & Cruz-Aldan, 1998); and La Sepultura Biosphere Reserve, Chiapas (n = 98) (Espinoza-Medinilla et al., 2004).

The great richness of species in La Chinantla is the result of the diversity of ecosystems found there, such as the Montane cloud forest, Tropical rainforest, and Pine-Oak Forest, which in most of our coverage were in a good state of conservation (Pennington & Sarukhán, 2005). Several factors may be contributing to this, such as the inaccessibility to certain areas, but above all, the conservation work that the residents of La Chinantla carry out through conservation zones, such as the ADVC (Acevedo, Lugo Espinosa & Ortiz Hernández, 2021). In addition to the strategies employed by the Chinantecos to safeguard their territory, they establish various rules of utilization and access within the framework of an internal regulation. These rules include restrictions on hunting, extraction of plant species, removal of vegetation cover for agricultural and livestock activities, among others (Anta-Fonseca & Mondragón-Galicia, 2006).

Species richness was higher in the lower zone (Zone 1), mainly comprising well-preserved Tropical Rainforest (TRF) and remnants of TRF with secondary vegetation (Pennington & Sarukhán, 2005; Meave et al., 2017). These vegetation types are well-represented within the 10 ADVCs present in the area, which likely contributes to the presence of many species. Zone 3 exhibited the second highest richness, despite having the lowest tree cover in the region (INEGI, 2016). However, there are numerous well-preserved TRF remnants and different successional stages interspersed with agricultural sites.

Bats had the greatest species richness, much of it in the lower zone. This could be explained by high coverage of tropical rainforest and fruit crops, where the species have been observed to find food and shelter in other tropical environments in America (Lira-Torres, Galindo-Leal & Briones-Salas, 2012; Briones-Salas, Lavariega & Lira-Torres, 2019). On the other hand, several species were recorded only in this area, some of which are considered “rare” such as Balantiopteryx io, Diclidurus albus, Micronycteris microtis, Mimon cozumelae, and Vampyrum spectrum. Some studies have shown that the presence of these species are good indicators of low habitat disturbance (Fenton et al., 1992; Wilson, Ascorra & Solari, 1996; Castro-Luna & Galindo-González, 2012). It is important to highlight that some bats in La Chinantla are also distributed in North America (e.g., Dasypterus mexicanus, D. xanthinus, and Myotis volans), South America (e.g., Balantiopteryx plicata, D. albus, and Saccopteryx bilineata), and Mesoamerica (e.g., B.io, Dermanura tolteca and Glossophaga leachii) (Villa & Cervantes, 2003). The area’s combination of tropical, semitropical, and temperate environments contributes to this high diversity of bats and has been observed elsewhere in Mexico (Briones-Salas & Sánchez-Cordero, 2004).

Small mammals exhibited the second highest species richness, dominated by species from the family Cricetidae, which can establish themselves in a wide range of environments, including disturbed sites (Linzey et al., 2012; Martin-Regalado et al., 2019). Another highly represented family in La Chinantla was Soricidae, with nine species. Four of these species were collected in sites with high humidity and good conservation status, primarily Pine-Oak Forest and Montane Cloud Forest, as documented by other authors (Carraway, 2007). Unlike bats, the highest species richness of small mammals was recorded in higher altitude zones. In fact, over half of the species ( n = 30) were exclusively found in these areas, including six species endemics to Oaxaca. Some authors have found high rodent diversity at elevated altitudes (Mccain, 2005; Martin-Regalado et al., 2019). It is worth mentioning that the six endemic species from Oaxaca, such as Habromys chinanteco and H. ixtlani, are found in sites protected by indigenous communities. The pattern observed for rodents is similar to that observed in the Andes of Peru and Chile, where speciation is proposed to have occurred in the highlands (Marquet, 1994).

Regarding medium and large mammals, most species were recorded in the lowland and warm areas of La Chinantla, although there was some overlap in confidence intervals between Zones 1, 2, and 3, indicating that their assemblages are very similar. Several species in Zone 1 were recorded in agricultural areas and secondary vegetation, such as Odocoileus virginianus, Procyon lotor, Nasua narica, and Herpailurus yagouaroundi, many of which are common and tolerant to disturbance (Briones-Salas, Lavariega & Lira-Torres, 2019). Similar findings have been described in the Istmo de Tehuantepec, Oaxaca, Mexico, in human-modified environments dominated by agricultural and livestock activities (Cortés-Marcial & Briones-Salas, 2014). It is important to mention that most of these records were made within VCAs by community monitors, reflecting the crucial role of indigenous communities in biodiversity knowledge and conservation.

The presence of all six species of felids distributed in Mexico is noteworthy, particularly Panthera onca, which has a comparable density to the Eden Reserve located in Yucatan and northern Sonora (Rosas-Rosas & Bender, 2012; Ávila Nájera et al., 2015; Lavariega et al., 2020). Meanwhile, although no population studies have been conducted on Puma concolor, the species has been recorded throughout La Chinantla (Figel et al., 2009; Silva-Magaña & Santos-Moreno, 2020). On the other hand, the bobcat, Lynx rufus, was found in the Pine-Oak Forest at the higher parts of the region. Another species of interest is Ateles geoffroyi, which was found in Zones 1 and 3 in the Tropical Rainforest and Montane Cloud Forest (Briones-Salas et al., 2006); even in the lowest zone, an ADVC called “Cerro Chango” was established due to the presence of groups of this species. Zone 3 had the lowest species richness, with only a few species recorded: the pocket gopher Orthogeomys grandis, the bobcat L. rufus, the hog-nosed skunk Spilogale angustifrons, and the ringtail Basariscus astutus.

Similarity indices

The greatest similarities between the four groups of mammals were recorded between neighboring areas. This indicates that there is an exchange of species between nearby zones. The analysis of beta diversity components indicates that the species turnover component was larger than nestedness, contributing to beta diversity in the Chinantla region. The intermediate zones may function as corridors that connect the upper and lower parts of La Chinantla. The landscape of the region presents a mosaic composed of Tropical Rainforest, Montane Cloud Forest, small areas of coffee plantations, cultivated areas, and pastures that may allow organisms move through.

Conservation

The Chinantecos play a crucial role as custodians of endangered species, as 33% of the total registered species are classified under some level of risk. For three of these species, La Chinantla is the northern limit of their distribution in the American continent with adequate habitat to maintain populations: the tapir Tapirus bairdii (Vivas-Lindo et al., 2020); the spider monkey A. geoffroyi (Vidal-García & Serio-Silva, 2011; Calixto-Pérez et al., 2018); and the spectral vampire V. spectrum, which finds refuge sites in the remnants of the region’s Tropical Rain Forest.

The study region is also a potential habitat for four medium-sized cats—H. yagouaroundi, Leopardus pardalis, Leoparuds weidii, and L. rufus—and for the last one, the zone represents its most southeastern distribution area (Monroy-Vilchis, Zarco-González & Zarco-González, 2019). This area has a substantial population of P. onca, the largest of these cats (Lavariega et al., 2020); it is also a biological corridor that connects populations in north of the country (Rodríguez-Soto et al., 2011).

La Chinantla region has a mosaic of landscapes that makes it heterogeneous in terms of vegetation, topography, and climate; this region also has high species richness and diversity and should be considered in state and national conservation policies. The establishment of 31 ADVCs by indigenous communities has significantly contributed to the conservation of mammals and other taxonomic groups. The Chinantecs who settle in the place have carried out community monitoring actions over the past 12 years, not only with mammals but other groups too such as birds (Ortega-Álvarez et al., 2012; Noria-Sánchez, Prisciliano-Vázquez & Patiño Islas, 2015), amphibians, and reptiles (Simón-Salvador et al., 2021). This has undoubtedly contributed to the knowledge and conservation of La Chinantla’s wild fauna.

Finally, the ADVCs in the region function as islands of conservation and protection for germplasm of the native fauna. We believe that strategies such as the establishment of biological corridors that connect these ADVCs will help enhancing the genetic exchange of populations and the dispersal of mammals in the area. Our study reveals that beta diversity is an important component in the region, as there is a high species turnover among different altitudinal gradients in this mountainous system. This could represent a first step towards identifying a suitable region for establishing an Archipelago Reserve, as proposed by other studies in Mexico (Moctezuma, Halffter & Arriaga-Jiménez, 2018; Garcia-Grajales & Buenrostro-Silva, 2009) and other Central American countries (Anderson & Ashe, 2000).

Archipelago Reserves also take a social approach that coincides with the Chinantecs’ strategies. This approach strengthen species conservation by using a community scheme through local and regional strategies. In addition, government programs have played an important role in channeling economic incentives through programs such as payment for environmental services and ADVCs certification, which make it possible to meet various conservation objectives. This would greatly support biodiversity, particularly for species that are endemic to the country and/or are classified in some threat category by national or international standards. Archipelago reserves are a novel and compelling proposal for biodiversity conservation as they seek to protect a set of complementary areas, such as the ADVCs in La Chinantla, which together safeguard an substantial portion of Mexican biodiversity. Considering the region’s high richness of mammals and other taxonomic groups, establishing protected areas by indigenous communities can be an alternative to the lack of officially designated protected areas.

Supplemental Information

Supplemental Information 1 Raw Data

Click here for additional data file.

Supplemental Information 2 Dates on which the camera traps were placed in the field in Chinantla

Click here for additional data file.

We thank N. Martín-Regalado for their comments on this manuscript. Special thanks to the researchers who allowed access to the scientific collection visited.

Additional Information and Declarations

Competing Interests

Author Contributions

Animal Ethics

Field Study Permissions

Data Availability

The authors declare there are no competing interests.

Miguel Briones-Salas conceived and designed the experiments, performed the experiments, analyzed the data, prepared figures and/or tables, authored or reviewed drafts of the article, and approved the final draft.

Rosa E Galindo-Aguilar conceived and designed the experiments, performed the experiments, analyzed the data, prepared figures and/or tables, authored or reviewed drafts of the article, and approved the final draft.

Graciela E González performed the experiments, authored or reviewed drafts of the article, and approved the final draft.

María Delfina Luna-Krauletz performed the experiments, authored or reviewed drafts of the article, and approved the final draft.

The following information was supplied relating to ethical approvals (i.e., approving body and any reference numbers):

Scientific collection issued by the Secretary of Environment and Natural Resources of Mexico (FAUT-0037; SEMARNAT).

The following information was supplied relating to field study approvals (i.e., approving body and any reference numbers):

The Comisión Nacional de Areas Naturales Protegidas approved the study (DRFSIPS-0095-2019).

The following information was supplied regarding data availability:

The raw data are available in the Supplemental Files.

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
