# Peer review of "Diversity and conservation of mammals in indigenous territories of southern Mexico: proposal for an “Archipelago Reserve”"

_PeerJ, doi:10.7717/peerj.16345_

## Round 0.1 · original submission · Major Revisions

The reviewers did a very good job of proofreading your article. They identified strengths in your paper but also points that have to be improved before your paper can be considered for publication in PeerJ.

Reviewer 1 ·

Basic reporting

The authors present an interesting study by collecting and compiling information on terrestrial mammals across an altitudinal gradient in southern Mexico. The study is important for summarizing information on a heterogeneous and biodiversity-rich region, which includes several species of conservation concern. Overall, the study is well-written and the results are thoroughly discussed. However, the methods section needs more attention; there is much information that needs to be included and explained. I have made comments and suggestions that I hope will help the authors to improve their manuscript quality. In addition, adjustments in the English language may benefit the manuscript.

Experimental design

L 116-125 – Is there any explanation or biological response for the altitude thresholds adopted here? It would be interesting to include some information. It would also be interesting to show (as supplementary material) the location of these ADVCs.
L 126 – ‘Obtaining records’, more information is needed here. For example, it would be informative to know how many areas you have sampled, the time interval of the sampling, which configuration was used in the camera traps, the sampling effort per method per area, the total sampling effort per method, and which species were recorded per area. Most of this information can be added as supplementary material, but I find it important to present them.
L 130-132 – Please include a reference or justification for the use of these body mass thresholds.
L 138-140 – Tomahawk traps were used to sample small mammals, correct? If yes, move this sentence together with methods to sample small mammals.
MS 1 – Which document this refers to?
L 153-155 – More information on the literature search needs to be included here. Which keywords were used? In Spanish and English? What were the criteria used to select studies? All studies found were included? It would be interesting to create a table with the list of studies selected to compose the database, and some information about them, such as group inventoried (volant, small, or medium and large-sized), number of species, etc.
L 156-163 – There is information missing here. These camera traps are the same mentioned in L 140-142? It would be interesting to depict the location of 18 areas sampled by camera traps. How many cameras were deployed per area? Which configuration was used? The sampling effort per area was the same?
Supplementary material 1 – Some of the information in this file is in Spanish, please correct them. Also, the references mentioned in the table need to be presented in the document.
L 172-173 – Did you calculate species accumulation curves for the entire region, zones, and vegetation type? If yes, describe it here. Also, you used the iNEXT R package, correct? Please provide references for both the package and R.
L 174-177 – More information about the comparisons needs to be included here. Which pairs of events/situations/assemblages were compared here?
L 179 – I suggest updating using the most recent list of IUCN (2022-2)

Validity of the findings

no comment

Additional comments

Introduction
L 61 – Please mention here some of these species.
L 80-82 – Please check this sentence, it lacks something.

Results
L 184 – What do these percentage values refer to? The different zones?
L 185 – Please call Table 1 here.
Table 1 – Please check the table caption, there is information missing on the acronyms mentioned in the cells. What does NOM mean?
L 186-198 – Although the information is available in Table 1, I think it would be informative for the readers to see this in a more visual form, e.g., the number of species per data collection method, and the number of species per size/class per vegetation type. I suggest using a multi-panel figure to show this (a bar chart, for example).
L 192-198 – This information is already synthesized in Table 1.
L 199-233 – There is much information that is repeated from the figures and tables, I suggest presenting only the most important ones, such as those that were statistically different (without overlap of confidence intervals). Also, Figure 3 appears before Figure 2, please their order.
Table 2 – I suggest presenting only two decimals for the values of the Jaccard index. The table caption mentions only three zones, while four were analyzed, please check.

Discussion
L 277-278 – Please check this sentence.
L 278-281 – What do you mean? High density?
L 316 – In Table 1 you mention Puma yagouaroundi. Please check.
L 333 - Tapirus

Reviewer 2 ·

Basic reporting

The paper is reasonably well-written and professional. However, the structure of the manuscript needs some work - there is a looseness in terms of the way some information is presented, which I highlight below. The literature is appropriate to the study, and for the most part, the background is sufficient.

There are some issues worth pointing out.

Abstract
1. Firstly, the Abstract is perhaps a little more loosely written than the rest of the paper - using terms like 'flyers' when you earlier said 'flying' was one of the categories (why not just call this category 'bats'? Indeed - you call them 'bats' in Figure 2 and Table 2, adding to the confusing range of terms).
2. Perhaps no need to introduce the named zones (e.g. 'Zone 4') here -it is somewhat distracting without some background/context. Perhaps just say something like "The greatest species richness was found at the highest elevations..." and then introduce the zone names in the methods and use them throughout the paper.
3. Similarly, you sau "the conservation proposals by indigenous people ..." but there is no context given for this in the Abstract - can you briefly describe here the lack of formal protected areas, and the proposals made by indigenous people?

Body of the Paper
Line 237: In some sections of the paper, species names are not italicised.
Line 255: remind the reader of MCF, TRF, and P-OF - perhaps better not to have abbreviations unless necessary because of frequent usage

Experimental design

See below under 'Validity of Findings' - I think the paper should be restructured to focus the paper more firmly on the issue or protected areas (PA) management. The stated aims at the end of the Introduction does not reflect an important focus on protected areas, and the ADVC's being an alternative approach to government PAs. The study could still examine species richness along altitudinal gradients, but the reason for doing so would become more apparent to the reader (and the study would have greater impact) - i.e., to better conserve biodiversity and to support indigenous protected areas management, we need to understand patterns of species richness in the region, and how the distribution of ADVC's captures biodiversity - or if not, how they could be better aligned to do so.

The work done seems rigorous and at an appropriate standard, but Methods could be described better - especially the field survey methods - how many nights for each trapping survey for example? How many nights total for mist-netting? How were species identified? What model of camera, and what settings were used (flash type, height from the ground to camera, how many photos per trigger, what constitutes an independent event, etc).

I like the way the data were analysed and presented - this seems sound, and appropriate.

Validity of the findings

This manuscript for the most part presents the results of a fauna survey. In that respect, it is not particularly novel or compelling. But, the data are interesting cast in terms of elevation and species richness - this is a little more interesting. But the most interesting aspect is the PA (or lack of PAs) and how the results can inform the reserving of sites under the indigenous ADVC model. So, perhaps the lack of a reserve system, and the indigenous management angle could be introduced earlier, and form the core of the paper - by this I mean, the authors could make the lack of a formal PA network the reason behind the study, with the fauna survey (and other data gathering) aimed at informing the process of prioritising reserve selection or conservation. This then allows the authors to discuss the value of such reserves in the context of their species richness... I instead read the paper to be a fauna survey simply because there wasn't much done in the area (which is valid) rather than grabbing the reader with the main message - that is - (1) the area is rich in biodiversity but poorly surveyed, (2) there are no PAs in the area and so the wildlife is at risk, (3) but there is an alternative PA approach proposed by local indigenous people that could be important for protecting biodiversity, and (4) we set out to assess the value of these PAs in terms of species richness, focusing on changes in richness within 4 elevation / forest zones, (50 our results will help the indigenous people in their conservation efforts... This sort of structure gives a better spin to the paper - putting up front the reason for the study.

Additional comments

None.

---

## Round 0.2 · Major Revisions

The reviewer has made a list of important, usefull corrections needed before your paper can be considered for publication and they annotated your manuscript. Please respond to each of their comments.

Reviewer 1 ·

Basic reporting

I commend the authors for the improvements made in the manuscript and for considering my and the other reviewer’s comments and suggestions. I have made more comments and suggestions to help improve the text and presentation of ideas, methods, and results. Please see them below. I have also made text suggestions and proposed some corrections in the attached files.

Experimental design

no comment

Validity of the findings

no comment

Additional comments

Introduction
L23 – I suggest removing ‘natural’.
L51 – One the largest humid tropical forest in the world? In Mexico? In the region?
L66-69 – I suggest highlighting that these species are threatened (at the regional or global level).
L70 – Include the scientific name for the jaguar.
L72 – If this region is important for jaguars as a corridor, I suggest replacing ‘jaguar populations’ with ‘jaguar conservation’ or adding the word ‘jaguar population maintenance’.
L74 – I suggest using protected area (PA) instead of protected natural area (PNA); the former is widely adopted in scientific literature and would be more easily indexed when people search for papers on this topic.
L75 – The ‘protected areas’ here, contradicts the previous sentence, which states that the region does not have any protected area. I suggest changing the sentence to something like: “However, Chinantecs have established almost 31 communitarian (or community-managed) conservation areas within their territory, which are volunteered elected for biodiversity conservation, namely…”.
L93 – What do you mean by ‘enhancing species turnover’? Related to beta diversity ‘species turnover and nestedness’? If yes, I suggest indicating ‘enhancing beta diversity’, which encompasses both processes.
L106 – And nestedness? Considering the concepts of beta diversity.
L107-109 – I suggest removing this sentence since it repeats some of the information in the previous and next sentences.

Results
L202-204 – Records from camera traps were obtained from CONANP? Is that correct? If so, I suggest rephrasing this sentence to make it clearer. Otherwise, you need to correct this sentence because it is not clear what you meant here.
Fig. S1 – Correct the title of the x-axis, it is in Spanish.
Table 1 – Here you separate medium from large-sized mammals, but in the methods, and across the text, you group them. I suggest defining the body mass threshold for large-sized mammals to make it clear for readers. Check commas between acronyms in the table, there are some typos and some of them missing. Also, double-check scientific names; species authority names are italicized and should not be.
Fig. 2 – I suggest organizing the panels horizontally since the y-axis has the same length (frequency). In addition, label them from A-C, which might facilitate indexing them in the text when you present the specific results for each category (zone, mammal class, vegetation). In the vegetation panel, for agriculture values, the numbers are overlapping.
L256-258 – I suggest presenting only the number of species for Felidae.
L259 – Lower altitudinal zone? If so, add ‘altitudinal’ to the sentence. The same for L261.
L264 – But statistically different only between Z1 and Z4, in which confidence intervals do not overlap.
L268 – The overlap was only between Z1 and Z4 for bats, compared to Z2 and Z3, it is statistically different.
L271 – All confidence intervals overlap for medium and large-sized mammals.
Fig. 3 – I suggest some improvements in this figure. First, label them from A-D, which might facilitate indexing them in the text when you present the specific results for each mammal category. You do not need to repeat the x and y-axis titles in all panels, you can align them in a way to suppress some of them. The same goes for the legend (zones and curves), which needs to be shown only once since the same colors and symbols are used in all panels.
Table 2 – Jaccard values are not italicized.
L275 – To facilitate comprehension, I suggest starting the sentence with “Considering all mammal species, 18 are shared…”.
L277-278 – Five species were exclusive of each zone? Not clear. The same for 278-279, you are talking about small mammals here? I suggest rewriting and merging the information in these sentences.
L280 – I suggest using ‘assemblage’ instead of ‘community’ when referring to the set of species recorded in each zone throughout the text.
Fig. 4 – Suggestion for the caption: “Jaccard dissimilarity among mammal assemblages considering all species, bats, small mammals, and medium and large-sized mammals in La Chinantla, Oaxaca”.
L293 – Call Table 1 at the end of this sentence.

Discussion
L329-331 – I suggest removing this sentence since it seems out of context with what is being discussed in this part.
L353 – I suggest defining what you consider micromammals and presenting references.
L368 – What SCA means?
L384-385 – I suggest removing this sentence since it repeats the results already presented.

Annotated reviews are not available for download in order to protect the identity of reviewers who chose to remain anonymous.

---

## Round 0.3 · Minor Revisions

Thank you for the deep work you have done for the consideration of the reviewer's comment. They have now checked your entire paper and made detailed minor recommendations, including via the manuscript annotation. Thank you for incorporating these recommendations into a revised version of your paper.

Reviewer 1 ·

Basic reporting

I commend the authors for the improvements in the manuscript. I have made some other suggestions and comments, please see below. I have made some text suggestions in the attached 'revised manuscript'.

Experimental design

no comment

Validity of the findings

no comment

Additional comments

Overall – Throughout the manuscript, you present diverse acronyms for vegetation, zones, protected areas, and others. However, their use is not standardized. Therefore, I suggest removing the acronyms from the text (except figures to save space) because sometimes they can be confusing (particularly for vegetation), and hard to remember their meaning.

Abstract
L29 – Disclose the body size thresholds used for small mammals and medium and large-sized ones.
L31 – I suggest replacing ‘community’ with ‘assemblage’ throughout the text.
L39 – I think it is not clear to the readers what an Archipelago Reserve means. I suggest explaining the term earlier and briefly in the abstract, replacing some of the information at the beginning.

Introduction
L86 – At national or international categories? Or both?
L100-113 – Merge these two paragraphs.
L111-113 – Conclusions.

Methods
L138 – I suggest replacing ‘Specimens’ with ‘Primary data collection’ since it also includes records from direct and indirect evidence of mammalian presence not including captures.
L171-172 – Which type of data was collected?
L195 – Did you search for the keywords in English and Spanish? If so, include this information in the text stating that you also search for the same keywords in Spanish.
L200 – Which authorities?
L220 – What do you mean by ‘traced’?

Results
L248 – It is not clear what these percentages refer to. Is it the different categories analyzed? Assemblage, small, bats, and medium and large-sized? If so, disclose them in the text.
Figure 2 – Since figures must be self-explicative, include the meaning of all acronyms in the figure caption. As you mention in the text the number of species for some of the categories presented in the panels, I suggest including above the bars, in bold, the total of number of species per category considered. After reading the text in L263-265, I observed that the colors do not match the number of species present in the text; please double-check this figure and the text.
L263-265 - After reading the text, I observed that the colors and numbers in Figure 2 do not match the number of species present in the text; please double-check this figure and the text.
L267 – I suggest standardizing the text concerning the use of ‘Zone’ and ‘Z’. Here, for example, you interchange both. Please check the entire manuscript.
L279-292 – I suggest calling Table 2 in the beginning since most of the information presented is presented there.

Discussion
L326-329 – I suggest including the size of these reserves for comparison.
L395-396 – I suggest removing this sentence because it has the potential to be misinterpreted and misused against ecosystem and species conservation.

Annotated reviews are not available for download in order to protect the identity of reviewers who chose to remain anonymous.

---

## Round 0.4 · accepted · Accept

I went through your manuscript and I thank you for having addressed the reviewer's remaining minor comments. The manuscript is now ready for publication.